# GPS Results from Long Time Monitoring of Geodynamic Processes in South-Western Bulgaria

**Nikolay Dimitrov** [1,*] and **Radoslav Nakov** [2]

1   National Institute of Geophysics, Geodesy and Geography, Bulgarian Academy of Sciences, 1113 Sofia, Bulgaria
2   Geological Institute "Strashimir Dimitrov", Bulgarian Academy of Sciences, 1113 Sofia, Bulgaria; radnac@geology.bas.bg
*   Correspondence: ndimitrov@geophys.bas.bg

**Abstract:** Monitoring of geodynamic processes by modern GNSS techniques in the area of Sofia and South-Western Bulgaria has continued for 25 years. To study the modern crustal movements in the area, Global Positioning System (GPS) data acquired between 1996 and 2021 are analyzed to obtain the velocity field for South-Western Bulgaria. For a time period of almost 25 years, the monitoring has covered 28 stations. They have been measured in different years and in a number of campaigns. Despite the difference in the measurements, the obtained results are quite homogeneous in the different localities of the studied area and show clear uniform tendencies. All velocities are in the southern direction. They are in the range of 1.5 mm/year to slightly over 3 mm/year, almost reaching 4 mm/year. The velocities of the stations tend to increase from north (stations around Sofia), passing through an intermediate locality (between Sofia and Kyustendil–Pazardhik), clearly increasing in the southernmost part of the country (around Gotse Delchev). This velocity field motivates N–S expressed extension with increasing rates from North to South. The difference in the velocity rates tends to change along geologically suggested active fault zones. The obtained results generally confirm previously data, but with much better accuracy and details at the local level. This way, both the repeated measurements and extension of the geodynamic network prove to be a powerful tool for a better understanding of present-day geodynamics.

**Keywords:** GPS; active tectonics; horizontal velocity; Bulgaria

## 1. Introduction

Tectonically, the territory of South–West Bulgaria is part of the South Balkan Extensional Region (encompassing South Bulgaria and Northern Greece), part of the broad East-Mediterranean–Balkan Extensional system [1]. Extension is the main form of deformation since the Middle Miocene time, being responsible for the occurrence of a differentiated topography. The latter is expressed by elevated mountains (horsts) and sedimentary basins (grabens) between them. Geological data point to a general N–S direction extension, resulting in structures with a general NW–SE to E–W trend. Deformation occurs along numerous active faults, as suggested by geological data. Historically, the area of Sofia is known for the occurrence of a few strong earthquakes, with suggested magnitudes of Mw 5.5–7.0 [2]. The latest one, of magnitude Mw 5.6, occurred on 22 May 2012 to the NW of Sofia around Pernik town [3] along a fault of NW–SE trend. On 4 April 1904, a strong earthquake of suggested magnitude Mw > 7 [4] occurred (to the south of Blagoevgrad, Figure 1). This event has been subject to numerous re-evaluations.

The location of South Bulgaria in a tectonically active zone with noticeable seismicity that predetermines the occurrence of dangerous geodynamic processes. These processes have the highest impact on the changes (deformations) of geodetic networks, built specifically for their study. Geodetic methods for estimating natural destructive processes provide specific quantitative values of recent crustal movements.

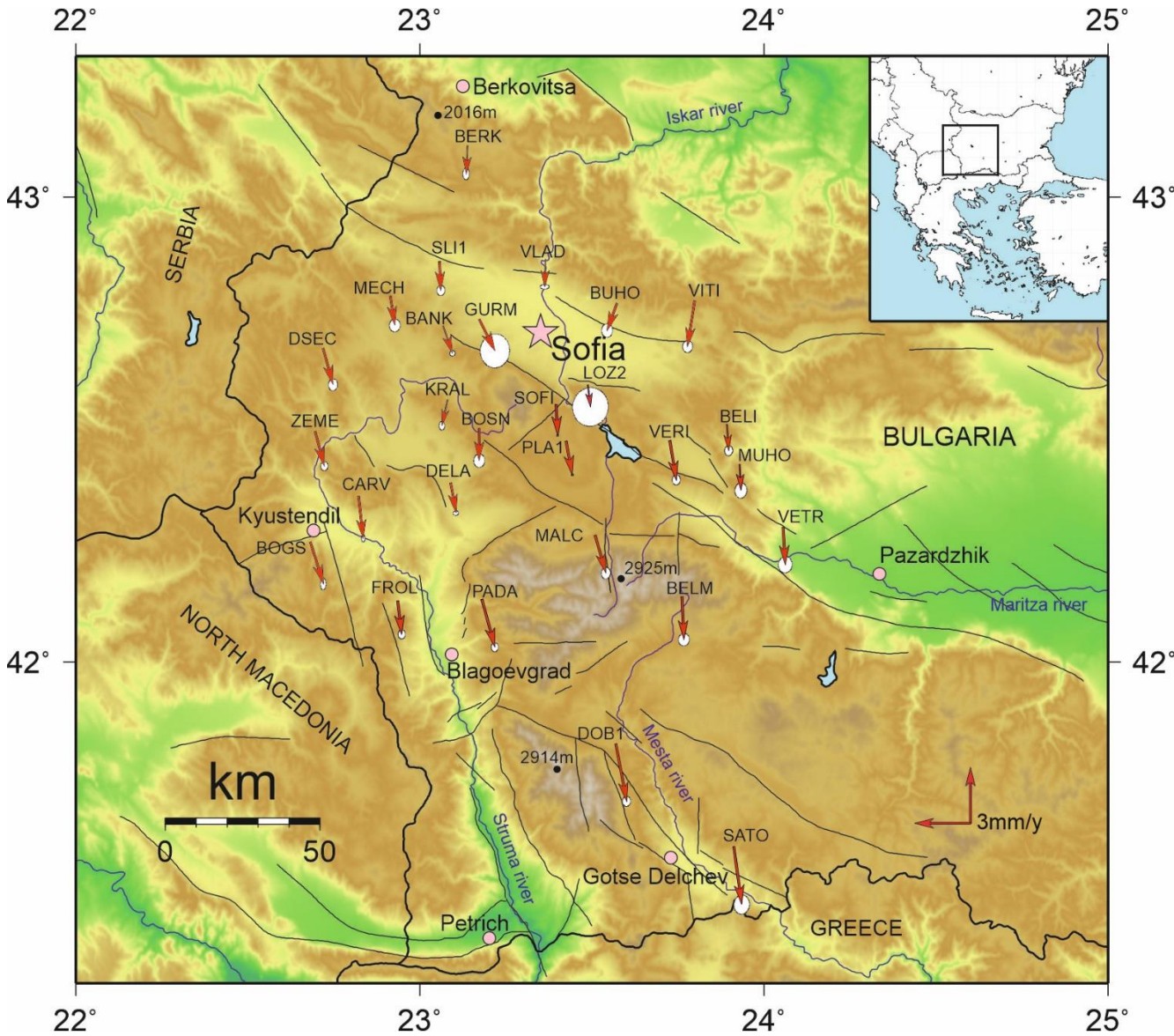

**Figure 1.** Horizontal GPS velocity with respect to Eurasian plate with 95% confidence. Slim lines show suggested active faults in the area. The topography is shown from green to brown—lower to higher elevation, respectively. The elevation of the highest peaks in the area is given in meters.

Monitoring of geodynamic processes by modern GNSS technics in the area of Sofia and South-Western Bulgaria has continued for 25 years. To study the modern movements of the Earth's crust during 1996–1997, a geodynamic network was built in the area around Sofia, covering South-Western Bulgaria. The network is designed for high-precision GNSS measurements, determination of coordinates and velocities of points, calculation of active strain in the area, and long-term monitoring crustal movements. The points have been stabilized so that the network covers the main tectonic structures in the area. The first GPS measurements of the Sofia Geodynamic Network were made in 1996. GPS measurement of all points of the network with processing and analysis of the results has, so far, been performed only in two epochs, in 1997 and 2000 [5–8]. During 2001–2020, the network was condensed and expanded with the stabilization of new points [9,10]. GPS measurements were periodically conducted for particular points of the network.

## 2. GPS Data and Estimation

At the end of 2019, the project "Monitoring of geodynamic processes in the region of Sofia" of the Department of Geodesy, the National Institute of Geophysics, Geodesy and Geography at BAS, funded by the Bulgarian National Research Fund, was launched [11]. This allowed for a new comprehensive measurement of the geodynamic network. The campaign was concluded in the summer of 2020 [10,12]. In the next summer, of 2021, three additional points were measured—BELM, SATO, and LOZ2 (Figure 1). Point BELM was previously measured in 1997 and 2000, and point SATO was measured in 1996 and 2003. The original point LOZE was destroyed after the measurements in 1997 and 2000; therefore, in 2021, we measured the duplicating point LOZ2, which was measured in 1997 but with a shorter observation period. For this reason, the obtained result for velocity of this point has greater error but is still reliable [13]. Point PLA1 is a CORS station in EPN Densification network from 2012 [14].

The measurements were processed/reprocessed in a two-step procedure using the GAMIT/GLOBK software v10.71 [15,16] to ensure the quality and homogeneity of the solutions. In the first step, loosely constrained estimates of station coordinates, Earth orientation, orbital parameters, and atmospheric zenith delays were determined using GAMIT. Major models and parameters used in GAMIT GPS data processing are given in Table 1. In the second step, a global Kalman filter was applied using GLOBK, to the combined, loosely constrained solutions and associated covariances, in order to estimate a consistent set of station coordinates and velocities. A six-parameter transformation was estimated by minimizing the horizontal velocities of 10 globally distributed IGS stations with respect to the IGS14 realization of the ITRF2014 reference frame [17]. The epochs of all GPS measurements included in this study is shown in Table 2.

**Table 1.** Major models and parameters used in GAMIT GPS data processing.

| | |
|---|---|
| Elevation cutoff angle and data weighting | 10°, data weighting depending on the elevation angle |
| Data sampling and data weighting | 30 s for data editing, and 120 s for parameter estimation |
| Antenna phase left | IGS ANTEX files are used to correct absolute PCVs of satellite and receiver |
| Ionospheric refraction | Ionosphere-free linear combination |
| Troposphere refraction | VMF1 for dry delay and parameter estimation in 2-h intervals for wet delays. Troposphere horizontal gradients in 24-h interval are estimated. Atmospheric tidal loading corrections VMF1 [18]. |
| Ocean tide | FES2004 model [19] with correction for the left-of-mass motion |
| Solid Earth tide, pole tide | Models recommended by IERS Conventions 2010 |

**Table 2.** Epochs of GPS measurements. The asterisk indicates the presence of measurements.

| Point ID | Year of Measurements | | | | | | | | | | | Number of Epochs |
|---|---|---|---|---|---|---|---|---|---|---|---|---|
| | 1996 | 1997 | 2000 | 2001 | 2002 | 2003 | 2004 | 2012 | 2017 | 2020 | 2021 | |
| BANK | | * | * | | | | | * | | * | | 4 |
| BELI | | * | * | | | | | | | * | | 3 |
| BELM | | * | * | | | | | | | | * | 3 |
| BERK | | * | * | * | | | | | | * | | 4 |
| BOGS | | * | * | * | | | * | | | * | | 5 |
| BOSN | | * | * | | | | * | | | * | | 5 |
| BUHO | | * | * | | | | | | | * | | 3 |
| CARV | | * | * | | | | * | | | * | | 4 |
| DELA | | * | * | | | | * | | | * | | 4 |
| DOB1 | * | * | * | * | | | | | | * | | 4 |
| DSEC | | * | * | | | | | | | * | | 3 |
| FROL | | * | * | | | | | | | * | | 3 |
| GURM | | | | | | * | | | | * | | 2 |
| KRAL | | * | * | | | | * | * | | * | | 5 |
| LOZ2 | | * | | | | | | | | | * | 2 |
| MALC | | * | * | | | | | | | * | | 3 |
| MECH | | | | | | * | | | | * | | 2 |
| MUHO | | * | * | | | | | | | * | | 3 |

**Table 2.** *Cont.*

| Point ID | Year of Measurements | | | | | | | | | | | Number of Epochs |
|---|---|---|---|---|---|---|---|---|---|---|---|---|
| | 1996 | 1997 | 2000 | 2001 | 2002 | 2003 | 2004 | 2012 | 2017 | 2020 | 2021 | |
| PADA | | * | * | | | | | | | * | | 3 |
| PLA1 | | * | * | * | | * | | * | * | * | * | 8 |
| SATO | * | | | | | * | | | | | * | 3 |
| SLI1 | | * | * | | | | | | | * | | 3 |
| SOFI | | * | * | * | * | * | * | * | * | * | * | 10 |
| VERI | | * | * | | | | | | | * | | 3 |
| VETR | | * | * | | | | | | | * | | 3 |
| VITI | | * | * | | | | | | | * | | 3 |
| VLAD | | * | * | * | | | | | | * | | 4 |
| ZEME | | * | * | | | | | | | * | | 3 |

## 3. Obtained Horizontal Velocities

The values of obtained velocities are shown in Table 3, where Ve and Vn are the east and north component of velocity in topocentric reference frame, respectively, and Sve and Svn represent the accuracy of the corresponding velocity component.

The 1-sigma quality of each station is 0.06–0.17 mm/year for NS component and 0.05–0.18 mm/year for EW. The exceptions are two of the stations (GURM and LOZ2), which have only two epochs of measurements for each, and, in both cases, the first epoch is shorter. Their 1-sigma qualities are greater and are between 0.37 mm/year and 0.42 mm/year, but still are reliable. When plotted with 95% confidence ellipses, the velocities of all stations fall within the bounds of the ellipses (Figure 1).

In the analysis, we examined the time series for all of the stations with measurements at three or more epochs, removing obvious outliers and further down-weighting those for which the normalized root-mean-square (nrms) was greater than 0.7. Five of these time series are shown in Figure 2. The rates and uncertainties estimated for the north and east components in the time series differ slightly from those given in Table 3 because, unlike the full velocity solution, the time series do not account rigorously for all correlations.

**Table 3.** Estimated horizontal velocities relative to Eurasia, ETRF2014.

| Point ID | Lat [°] | Long [°] | Ve [mm/year] | SVe [mm/year] | Vn [mm/year] | SVn [mm/year] |
|---|---|---|---|---|---|---|
| BANK | 42.72 | 23.07 | 0.60 | 0.06 | −1.65 | 0.06 |
| BELI | 42.51 | 23.89 | 0.18 | 0.11 | −1.53 | 0.09 |
| BELM | 42.14 | 23.76 | 0.10 | −2.29 | 0.12 | 0.14 |
| BERK | 43.11 | 23.14 | 0.05 | 0.13 | −1.61 | 0.07 |
| BOGS | 42.26 | 22.68 | 0.77 | 0.13 | −2.37 | 0.06 |
| BOSN | 42.50 | 23.17 | 0.09 | 0.15 | −1.88 | 0.12 |
| BUHO | 42.77 | 23.57 | −0.52 | 0.15 | −1.60 | 0.12 |
| CARV | 42.36 | 22.82 | 0.36 | 0.08 | −2.45 | 0.04 |
| DELA | 42.39 | 23.09 | 0.34 | 0.05 | −1.76 | 0.06 |
| DOB1 | 41.82 | 23.57 | 0.67 | 0.11 | −3.25 | 0.09 |
| DSEC | 42.68 | 22.72 | 0.60 | 0.13 | −2.09 | 0.10 |
| FROL | 42.13 | 22.94 | 0.23 | 0.10 | −1.92 | 0.08 |
| GURM | 42.74 | 23.17 | 1.07 | 0.37 | −1.80 | 0.32 |
| KRAL | 42.57 | 23.08 | −0.19 | 0.10 | −1.54 | 0.06 |
| LOZ2 | 42.60 | 23.49 | 0.15 | −1.18 | 0.39 | 0.42 |
| MALC | 42.27 | 23.51 | 0.61 | 0.13 | −2.14 | 0.10 |
| MECH | 42.79 | 22.91 | 0.40 | 0.14 | −1.74 | 0.12 |
| MUHO | 42.43 | 23.93 | 0.11 | 0.17 | −1.52 | 0.13 |
| PADA | 42.14 | 23.18 | 0.75 | 0.10 | −2.67 | 0.08 |
| PLA1 | 42.48 | 23.43 | 0.43 | 0.02 | −1.97 | 0.02 |
| SATO | 41.59 | 23.91 | 0.42 | 0.17 | −3.08 | 0.17 |
| SLI1 | 42.86 | 23.06 | 0.12 | 0.11 | −1.69 | 0.09 |
| SOFI | 42.56 | 23.39 | 0.11 | 0.02 | −1.87 | 0.02 |

**Table 3.** *Cont.*

| Point | Lat | Long | Ve | SVe | Vn | SVn |
|---|---|---|---|---|---|---|
| ID | [°] | [°] | [mm/year] | [mm/year] | [mm/year] | [mm/year] |
| VERI | 42.48 | 23.73 | 0.49 | 0.11 | −2.28 | 0.09 |
| VETR | 42.29 | 24.06 | 0.22 | 0.18 | −2.15 | 0.15 |
| VITI | 42.78 | 23.80 | −0.34 | 0.13 | −2.53 | 0.11 |
| VLAD | 42.87 | 23.36 | 0.12 | 0.05 | −1.60 | 0.10 |
| ZEME | 42.50 | 22.70 | 0.61 | 0.10 | −2.03 | 0.08 |

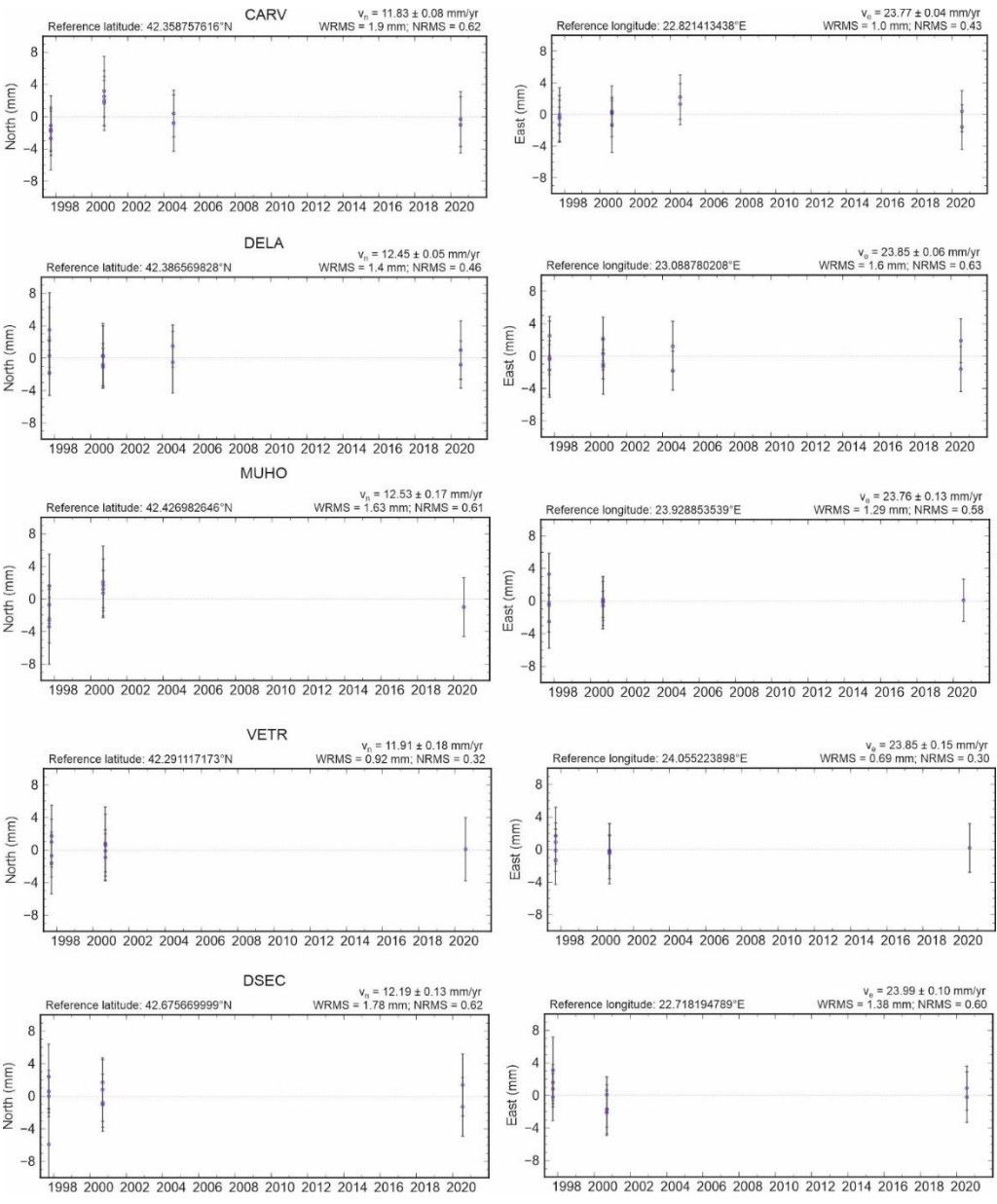

**Figure 2.** Long-term repeatability of the horizontal station positions.

All obtained horizontal velocities clearly exceed 1.5–2 mm/year, reaching up to 3–4 mm/year. As a general tendency, the velocities tend to increase from north to south. The northernmost area (N–NW of Sofia) sees a majority of velocities of about 1.5 mm (stations BERK, VLAD, BUHO, and SLIV). In the area to the south of Sofia, velocities of 2–3 mm/year (stations DSEC, CARV, BOGO, VERI, PADA, VETR, and MALA) dominate. Further south, the velocities of DOB1 and SATO are almost 4 mm/year, which is the high-

est in the area. The behavior of some stations with velocities from earlier campaigns [6] (e.g., VLAD and BUHO), that were showing significant deviation now fit in the general trend. This shows that the reason was the lack of accuracy due to the short period between observations, but not to tectonic reasons.

## 4. Comparison of the Results with Previous Studies

To evaluate the result, we combined the results for the obtained horizontal velocities of points from Sofia's geodynamic network with the results for the point velocities from the EPN densification project. EPN Densification is a joint venture of agencies and institutions from European countries, which operates and/or performs regular processing of the data from dense national GNSS networks. The primary goal of EPN Densification is to realize a continental-scale, homogeneous, high-quality position and velocity product. The generated results are extremely reliable and freely available for the geosciences community [20]. Figure 3 shows the horizontal velocities of points from Sofia geodynamic network along with the results for the point velocities from EPN densification project.

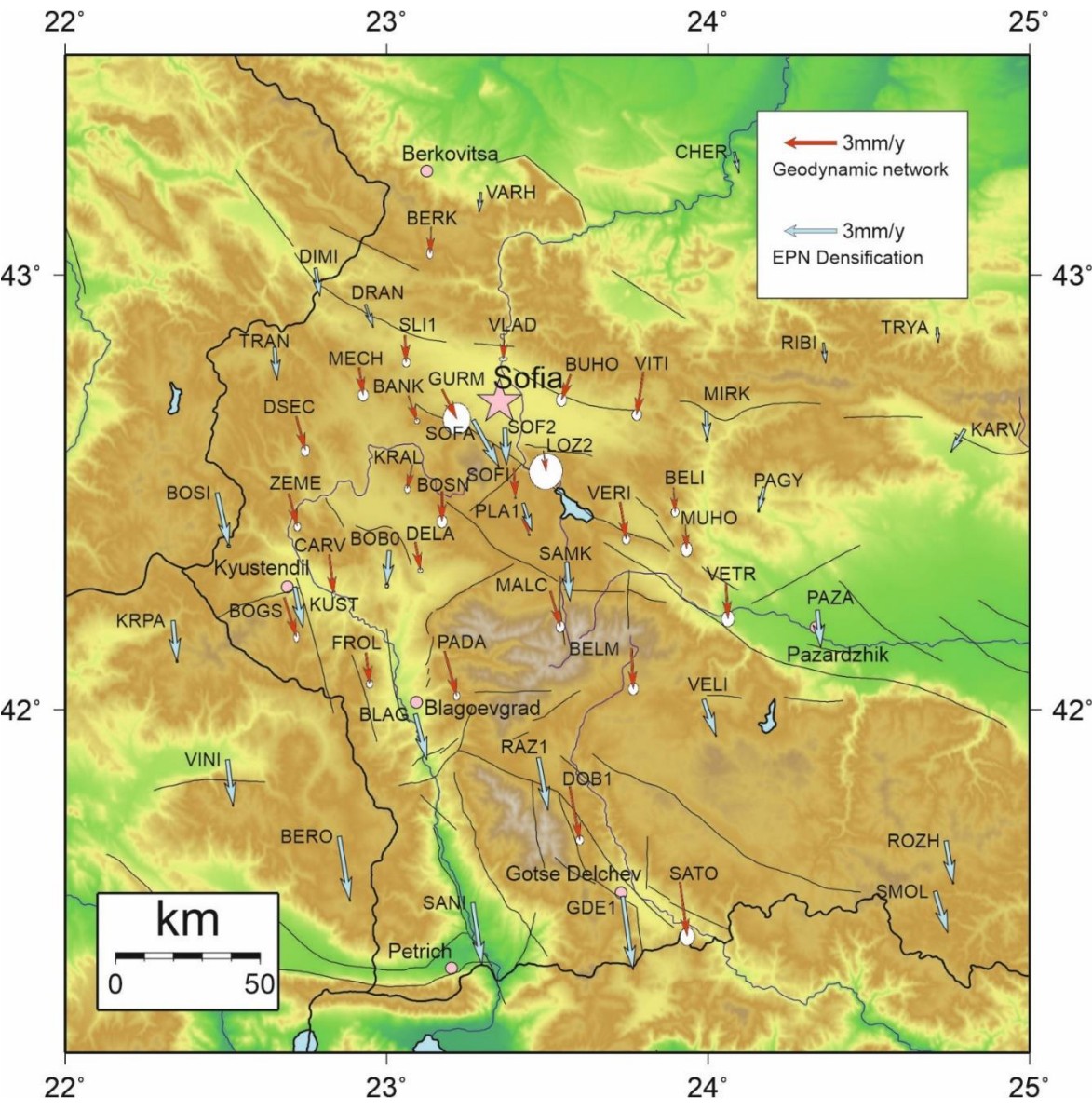

**Figure 3.** Horizontal velocities of the point from Sofia geodynamic network (red arrow) and from the EPN densification (light blue arrow).

Previously deployed EPN stations (Figure 3) cover a broader area, whereas our network is covering the area of SW Bulgaria, focusing on the territory with the densest population in Bulgaria—around Sofia. Our network is especially aiming to provide new, reliable data on present-day geodynamics, that may be used in the evaluation of the geological risk like seismicity, landslides, debris-flow, landslides, etc. Our results, compared to the results of EPN that cover the same area, are, in general, quite comparable. However, in some areas, some differences are seen. This is especially observed in the points south of Sofia, where the results from the previous measurements are not uniform with our data. This concerns, especially, station SOFA. At the same time, station PLA1 has almost a full concurrence of the results. As seen in this nodal locality, only a few stations were available, whereas in the results of the present study, a dense network of stations is now available. The results are very homogeneous and provide a reliable base for further studies.

The results of both networks supplement each other and are a base to evaluate the reliability of obtained velocities. From this point of view, the compliance and reliability are very high.

The comparison of velocities from the two different networks allows some important conclusions on the role of some faults to be made, especially where EPN points cover bordering areas not covered by our observations. This is obvious in the SE part of the studied area (Figure 3). The sharp difference between stations RAZ1, DOB1, GDE1, and SATO from the west and stations SMOL and ROZH from the east clearly demonstrates the activity of the NW–SE trending fault, known as the Dospat fault. To the west, is formed an area with well-expressed higher velocities moving to the south. The difference in velocities suggests a significant amount of extension along this fault.

In the NE part of the area, we do not have our own measurements. The velocities of the points of EPN (RIBI, TRYA, MIRK, and KARV) tend to show visibly lower velocities compared to our westerly located points (VITI, VERY, and MUHO). It remains unclear if these results are due to network systematic errors or whether they reflect geodynamic differences.

## 5. Discussion and Geodynamic Implications

For a time period of almost 25 years, the monitoring has covered 28 stations. They have been measured in different years and in a different number of campaigns (Table 2). The number of campaigns varies from two (stations GURM, LOZE2, and MECH); three—fourteen stations; four—six stations; five—three stations; eight—one station (PLA1); and ten—1 station (SOF1). Despite the different number of the measurements, the obtained results are quite homogeneous in the different localities of the studied area and show clear uniform tendencies. In relation to that, the velocities of stations PLA1 and SOF1, with the highest number of measurements, which suggests a very high reliability, do not differ significantly from surrounding stations. This is quite obvious when compared to the very close station LOZ2. The latter, having been measured only twice, shows a slightly slower velocity, but still in the same range and direction. Station MECH (two epochs of measurements) also fits in the local pictures of the surrounding stations (DSEC, BANK, and SLI1). The station GURM shows some slight local deviation, which may be due to the only two epochs of measurement, but may also reflect some specific tectonic setting. To solve this problem, further measurements are needed. A general conclusion that could be made is that the different number of measurements do not affect the significance of the obtained results. However, the greater numbers of measurements provide a higher reliability.

Some general remarks may be made:

All velocities are in a southern direction. They are in the range of 1.5 mm/year to slightly over 3 mm/year, almost reaching 4 mm/year (stations DOB1, SATO). The velocities of the stations tend to increase from north (stations around Sofia) to south, passing through an intermediate locality (between Sofia and Kyustendil–Pazardhik), clearly increasing in the southernmost part of the country (around Gotse Delchev). In general, the velocities are oblique to the faults. This is better expressed in the northern part (stations SLIV, MECH, BANK, GURM, BOSN, and VERI), suggesting oblique extension to the structures. Instead,

in the SW part (stations ZEME, CARV, FROL, and PADA) the velocities are parallel to some faults, suggesting strike-slip deformation. North of station BOGO extension is probably on the northern located fault (Kustendil fault) and strike-slip deformation to the eastern fault [10].

This velocity field motivates N–S expressed extension with increasing rates from North to South. The difference in the velocity rates tends to change along geologically suggested active fault zones. This is quite obvious between stations PADA, MALC, and BELM from the south and northern lying stations (DELA, PLA1, MUHO, VETR, etc.). This result points to the significance of the GPS monitoring for the identification and evaluation of active faults.

### 6. Conclusions

The presented data result from two campaigns made in 1997 and 2000, that are enlarged by a third campaign from 2020 and supplemented by an extension and acquirement of new data in 2021. The results provide new data on a tectonically and seismically active area. Compared to the first published results of the network [5,6,21], the present results confirm that the general tendency of movement of the stations in the region of Central West Bulgaria is in the south direction with respect to stable Eurasia.

The velocities are in the range of 1.5–2 mm/year to 3–4 mm/year. They increase generally from north to south. The change in the velocity field tends to be related to active faults. Extension is shown to be the main mechanism of deformation.

The obtained results in a general way confirm previously data, but with much better accuracy and details at local level. This way, both the repeated measurements and extension of the geodynamic network prove to be a powerful tool for better understanding present-day geodynamics.

**Author Contributions:** Conceptualization, N.D.; Investigation, N.D. and R.N.; Writing—original draft, N.D. and R.N. All authors have read and agreed to the published version of the manuscript.

**Funding:** This research was funded by National Science Fund, Bulgaria, Contract No КП-06-Н 34/1, project "Monitoring of geodynamic processes in the area of Sofia". Call identifier "Competition for financial support of basic research projects—2019".

**Institutional Review Board Statement:** Not applicable.

**Informed Consent Statement:** Not applicable.

**Data Availability Statement:** Not applicable.

**Acknowledgments:** We are grateful to the staff of the project "Monitoring of geodynamic processes in the area of Sofia" for their hard work. We obtained global solution files and most of the IGS tracking data from the Scripps Orbit and Permanent Array Center (SOPAC) and Crustal Dynamics Data In-formation System (CDDIS). We thank Robert W. King, for helpful advice and discussions. The maps in this paper were generated using the public domain Generic Mapping Tools (GMT) software [22]. For topography we used data from the General Bathymetric Chart of the Oceans (GEBCO). We thank three anonymous reviewers for comments that have improved the manuscript.

**Conflicts of Interest:** The authors declare no conflict of interest.

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
