# Peer review of "GPS Results from Long Time Monitoring of Geodynamic Processes in South-Western Bulgaria"

_applsci, doi:10.3390/app12052682_

Round 1

Reviewer 1 Report

Thank you for the interesting work. The paper presents GPS results from long time monitoring of geodynamic processes in South Western Bulgaria. In general, it is well structured and easy to follow. However, the contribution needs improvements before having a chance of being accepted for publication.

  1. The papar is quite short for an article. It is not a problem. But a lot of detail is missing. For example, what is the detail processing strategy of GPS? Which models have been applied for GPS errors? Have you tried differents models to achieve the best accuarcy?
  2. It would be nice to present Table 1 with a figure. 
  3. The paper confirms the previously published results with new data. However, it is not clear how much improvements have been obtained. And what is the contribution (or Novelty) of the paper? please be more specific.

Author Response

Thank You for your review.
We have accepted most of your comments. 
We added a table with major models and parameters used in GAMIT GPS data processing. We added text (in point 3) explaining the high accuracy of the horizontal velocities obtained as a result. We consider this to be a very important result,  which has not been the case so far. In the text below we have added an explanation of the confirmations and the differences compared to previous studies.
A figure can be placed with Table 1, but it will contain the same information as the other figure, so we think it is better not to place another figure. 

Reviewer 2 Report

The article is well written and can contribute in solving the most important geodynamics and geophysics problems.

I have some question and recommendations to the authors.

1) How can  be implemented the results of the investigation presented in the paper  in geophysics and geodesy? First of all, I mean the interpretation problems of gravity and magnetometry and also the earthquake prediction. 

2) Can the authors describe the velocity fields evolution in the region under investigation?

3) Some comparison analysis with the results obtained by other researchers should be fulfilled in order to reveal the main trends in geodynamics of the region under consideration.

Author Response

Thank You for your review.
We have accepted most of your comments.
We added a new point 4. Comparison of the results with previous studies.
In this article we examine the geodynamic situation in the area.
The results of the investigation presented in the paper can be used 
by other scientists in field of geophysics, magnetometry and seismology
in their research. 

Reviewer 3 Report

The research is too simple, authors have to make analysis more about the velocity of the movement, why it occurred, make relation with other sensors, geology map, the structure of the soil, relation with earthquake if it occurred. Then, please make it a good conclusion. The paper has to improve then resubmit it again.

Author Response

In this article we examine the GPS velocities and their effect on the geodynamic situation in the area. 
We especially focus on the obtained results and point on the long term observations and their methodological value.
We added text (in point 3) explaining the high accuracy of the horizontal velocities
obtained as a result. We consider this to be a very important result,
which has not been the case so far.
We added a new point 4. Comparison analysis of the results with previous studies.
The results of the investigation presented in the paper can be used
by other scientists in field of geophysics, active tectonics, seismology, etc.,
in their research. Probably, we do not need a geological map as the basement rocks have little to do with present day geodynamics. 
Our figures show the best expressed active faults. Further investigations related to seismicity are foreseen. 
In this paper we notice, that this is an active seismic area.

Round 2

Reviewer 1 Report

Thank you for the reply.

Author Response

Thank You for your review.

Reviewer 3 Report

Thank you very much for the improvement of the manuscript. Before the publication, please add :

  1. the graphic of time series in x, y coordinate of table 2. It is easier to read to check the result, where the jump and why it occurred. Please make difference between, x position only, y position only, and z. also the acceleration

    2. Please add also a different every year in point 1 (1996-2021), and show it using a graphic of time series, then make the sign of anomalies, add analysis of it. It is very important to check the anomalies using graphs of time series every year, or authors could use one-time series of graphic.

Author Response

Thank You for your review.
We add additional text to explain the using of time series
and a picture with graphs of 5 of the station timeseries for example.

Regards.